# Osteoporosis Preclinical Research: A Systematic Review on Comparative Studies Using Ovariectomized Sheep

**DOI:** 10.3390/ijms23168904

**Published:** 2022-08-10

**Authors:** Francesca Salamanna, Deyanira Contartese, Francesca Veronesi, Lucia Martini, Milena Fini

**Affiliations:** 1Complex Structure Surgical Sciences and Technologies, IRCCS Istituto Ortopedico Rizzoli, 40136 Bologna, Italy; 2Scientific Direction, IRCCS Istituto Ortopedico Rizzoli, 40136 Bologna, Italy

**Keywords:** osteoporosis, ovariectomy, bone remodeling, sheep, systematic review

## Abstract

Sheep ovariectomy (OVX) alone or associated to steroid therapy, deficient diet, or hypothalamic–pituitary disconnection has proven to be of critical importance for osteoporosis research in orthopedics. However, the impact of specific variables, such as breed, age, diet, time after OVX, and other variables, should be monitored. Thus, the design of comparative studies is mandatory to minimize the impact of these variables or to recognize the presence of unwanted variables as well as to better characterize bone remodeling in this model. Herein, we conducted a systematic review of the last 10 years on PubMed, Scopus, and Web of Knowledge considering only studies on OVX sheep where a control group was present. Of the 123 records screened, 18 studies were included and analyzed. Results showed that (i) Merino sheep are the most exploited breed; (ii) 5–6 years of age is the most used time for inducing OVX; (iii) ventral midline laparotomy is the most common approach to induce OVX; (iv) OVX associated to steroid therapy is the most widely used osteoporosis model; and (v) success of OVX was mostly verified 12 months after surgery. In detail, starting from 12 months after OVX a significant decline in bone mineral density and in microarchitectural bone parameters as well as in biochemical markers were detected in all studies in comparison to control groups. Bone alteration was also site-specific on a pattern as follows: lumbar vertebra, femoral neck, and ribs. Before 12 months from OVX and starting from 3–5 months, microarchitectural bone changes and biochemical marker alterations were present when osteoporosis was induced by OVX associated to steroid therapy. In conclusion, OVX in sheep influence bone metabolism causing pronounced systemic bone loss and structural deterioration comparable to the situation found in osteoporosis patients. Data for treating osteoporosis patients are based not only on good planning and study design but also on a correct animal use that, as suggested by 3Rs principles and by ARRIVE guidelines, includes the use of control groups to be directly contrasted with the experimental group.

## 1. Introduction

Osteoporosis is a chronic disease typified by bone loss and increased skeletal fragility; it is associated with decreased quality of life, loss in mobility, and chronic pain [1]. Globally, it is estimated that osteoporosis affects 200 million women worldwide. A total of 1 in 3 women over age 50 will undergo osteoporosis -related fractures, as will 1 in 5 men aged over 50 [2,3]. Osteoporosis has several etiologies, but the most common cause is estrogen deficiency-related bone loss that occurs after menopause [2,3,4]. Since its underlying regulatory mechanisms are still not fully understood and treatment strategies are not adequately resolved, there is a great need for osteoporosis research associated to estrogen deficiency-related bone loss [4]. For this research, animal models are critical and recommended by the Food and Drug Administration (FDA) and by World Health Organization (WHO). Several animal models for postmenopausal osteoporosis and disuse osteoporosis, as well as glucocorticoid-induced osteoporosis, are described in the literature, and their phenotypes seem to mimic osteoporosis in humans (Figure 1) [5]. One of the most common animal models for postmenopausal osteoporosis is generated in mouse, rat, sheep, and nonhuman primates by ovariectomy, the surgical removal of one or both ovaries (Figure 2) [5,6,7]. In ovariectomized (OVX) rodents, the opportunity to alter/modify the genetic background offers the chance to specifically characterize the role of individual gene products in bone metabolism and/or bone disease [6]. However, specific aspects, such as fracture healing and orthopedic implant osteointegration, can be advantageously managed with large animal models [6]. Based on the literature data, sheep have proven invaluable in orthopedic research and specifically for osteoporosis research [6,7]. In addition to the macro- and microarchitecture of bone, sheep bone is comparable to human bone. OVX in sheep is a simple and safe surgical approach, and sheep also have easy husbandry needs and a compliant nature. Although some groups questioned sheep’s bone mineral density (BMD) loss after OVX and observed a rebound effect by histomorphometrical analyses [8,9,10,11], most groups reported significant changes in bone mass and micro-structural parameters, as well as biomechanical properties, from 6 to 24 months after OVX [12,13,14,15,16]. The significance of OVX effects on bone metabolism in sheep is also strengthened by several studies demonstrating that estrogen and selective estrogen receptor modulators (SERM’s) can significantly increase bone mass in these animals [17,18]. Furthermore, in Dorset and Merino sheep, the estrous cycle is longer than that of any other breed. The reproductive cycle during the breeding season (10 months) is strikingly similar to that of women, with an estrous cycle of 14 to 20 days, averaging 16.5 days. The different findings between the studies may be due to a plethora of variables that can potentially affect the outcome. Results should always be interpreted against the background of breed, age, diet, skeletal site, time after OVX, and duration of study. Consequently, appropriate control groups are mandatory to minimize the impact of these variables.

Normally, each experiment should use control groups (negative, positive, sham, vehicle, and comparative) of animals that are contrasted directly to the experimental group, as also suggested by the 3Rs (replacement, reduction, and refinement) principles and by Animal Research: Reporting In Vivo Experiments (ARRIVE) guidelines. Thus, to better understand and characterize preclinical and translational models on altered bone metabolism due to OVX in sheep, we conducted a systematic review of the last 10 years of the literature, considering only studies where a control group was present. The presence of a control group allows us to have scientifically valid and reproducible data, which should be the main goal of any scientific investigation.

## 2. Materials and Methods

### 2.1. Eligibility Criteria

The PICOS model (population, intervention, comparison, outcomes, study design) was used to design this study: (1) studies that considered female sheep (population) submitted to (2) OVX-induced estrogen deficiency (interventions) (3) compared to animals on which a sham operation was performed or to animals with no operation (comparisons) (4) that characterized bone metabolism, its changes, and its enhancing after specific treatments (outcomes) in in vivo studies (5) (study design). Studies from 6 June 2012 to 6 June 2022 were included in this review if they met the PICOS criteria. We excluded studies on the use of specific anti-osteoporotic therapies and biomaterials, on male animals as well as studies involving female animals with additional pathological conditions. Additionally, we did not consider human, in vitro, ex vivo, or in silico studies, editorials, reviews, systematic reviews, and meta-analyses.

### 2.2. Search Strategies

Our literature review involved a systematic search conducted in June 2022. We performed our review according to the Preferred Reporting Items for Systematic Reviews and Meta-Analyses (PRISMA) statement [19]. The search was performed on three databases: PubMed, Scopus, and Web of Science Core Collection. The following combination of terms was used (ovariectomy) AND (sheep) AND (bone), and for each of these terms, free words and controlled vocabulary specific to each bibliographic database were combined using the operator “OR”. The combination of free-vocabulary and/or medical subject headings (MeSH) terms for the identification of studies in PubMed, Scopus, and Web of Science Core Collection were used (Table 1).

### 2.3. Selection Process

After the articles were submitted to a public reference manager (Mendeley Desktop 1.19.8) to eliminate duplicates, possible relevant articles were screened using title and abstract by three reviewers (FS, DC, and FV). Studies that did not meet the inclusion criteria were excluded from review, and any disagreement was resolved through discussion until a consensus was reached or with the involvement of a fourth reviewer (MF). Subsequently, the remaining studies were included in the final stage of data extraction.

### 2.4. Data Collection Process and Synthesis Methods

The data extraction and synthesis process started with cataloguing the studies’ details. To increase validity and avoid omitting potentially findings for the synthesis, three authors (FS, DC, and FV) extracted and performed tables, taking into consideration reference, objectives, study design, animal number and age, time from OVX, experimental time, analyses and measurements, and main results (Table 2).

### 2.5. Assessment of Methodological Quality

A quality assessment of all selected full-text articles was performed according to the ARRIVE guidelines for reporting in vivo experiments in animal research [20]. This guideline consists of a checklist of 20 items and has been developed using the CONSORT statement as their foundation [21,22]. Quality assessment was performed in two different phases. During phase I, quality assessment was based on the published full-text article performed independently by the three authors (FS, DC, and FV). In phase II, disagreements were resolved by discussion.

**Table 2 ijms-23-08904-t002:** Studies’ characteristics.

Ref.	Objectives	Study Design (Groups)	Animal Number and Age	OVX (Time from OVX)	Experimental Time	Analyses and Measuraments	Main Results
Brennan et al., 2012 [23]	Evaluation of the timing of changes in bone	- Control (intact); - OVX	*n* = 19 mixed-breed 4 years	NR	12 and 31 mo after OVX	- Micro-CT: proximal femur BV/TV, Tb.N., Tb.Th, Tb.Sp; - Biomechanics: trabecular bone ultimate compressive strength; - RT-PCR: right metacarpal RANKL, OPG, COL1A1, COL1A2, OCN, OPN; - FTIR: mineral-to-matrix ratio, Crystallinity, Collagen crosslinking	OVX induced: ↑ RANKL, OPG, COL1A1, COL1A2, OCN, OPN by 12 and 32 mo; ↓ mineral-to-matrix ratio, ratio of mature to immature collagen cross-linking than Control at 31 mo; =Tb.N., Tb.Th, Tb.Sp by 12 mo; ↓ Tb.N, Tb.Th and ↑ Tb.Sp at 31 mo; ↓ compressive strength at 31 mo
Oheim et al., 2013 [24]	Relevance of low turnover OP induced by OVX and HPD	- Control (intact) 24 mo; - OVX 24 mo; - HPD + OVX 12 mo; - HPD + OVX 24 mo	*n* = 20 Corriedale 6–7 years	OVX performed 1 week before HPD	12 and 24 mo after HPD	- Biochemistry urine (deoxypyridinoline) and serum parameters (sodium, potassium, chloride, Ca, P, Crea, BAP) - X-rays, histology: iliac crest, spines (L3–L5) - histomorphometry (BV/TV, Tb.N, Tb.Sp, Tb.Th, Ob/B.Pm, N.Oc/B.Pm, Ob.S/BS and Oc.S/BS, ES/BS, OS/BS, MS/BS, BFR/BS - HR-pQCT: BV/TV, Tb.N, Tb.Sp, Tb.Th, Ct.Th - qBEI: BMDD - biomechanical test: femora three-point-bending test	OVX + HPD lead to a persisting low turnover status with negative turnover balance in sheep followed by cortical and trabecular bone loss with biomechanical impairment
Zhang et al., 2014 [25]	Variation of cancellous bones at four skeletal sites: lumbar vertebra, femoral neck, mandibular angle, and rib	- Control (intact) - OVX	*n* = 16 4 ± 0.5 years	12 mo after OVX	12 mo after OVX	- X-ray absorptiometry: lumbar vertebra BMD - micro-CT: BV/TV, Tb.N, Tb.Sp, Tb.N - histology - histomorphometry: Tb.Ar/T.Ar - biomechanical test: compression test	The sensibility of cancellous bones in OVX sheep was site-specific on a pattern as follows: lumbar vertebra, femoral neck, mandibular angle (↓ BV/TV by 45.6%, 36.1% 21.3% and 18.7% in lumbar vertebrae, femoral necks, mandibular angles, and ribs; Tb.N have the same downtrend; ↓ Tb.Ar/T.Ar by 32.1%, 23.2% and 20.7% in lumbar vertebrae, femoral necks, and mandibular angles)
Kreipke et al., 2014 [26]	Changes in microarchitectural and mechanical parameters in femoral condyles and vertebral bodies	- Control (intact) - OVX-1 (euthanized at 1 year) - OVX-2 group (euthanized at 2 years)	*n* = 19	1 or 2 years following the OVX	2 years	micro-CT (BV/TV, SMI, Tb.Th., Tb.Sp., BMD, TMD, DA)	↓ BV/TV, Tb.Th., BMD, ↑ SMI, Tb.Sp. in OVX-1 and OVX-2 vs control. ↑ mechanical anisotropy in OVX groups. OVX had minimal effects on trabecular architecture of the distal femur even after 2 years. Medial condyle: ↑ BV/TV, BMD, TMD in OVX-1 group vs. control. Lateral condyle: ↓ DA in OVX-1 group vs control. ↑ TMD in OVX groups vs control
Andreasen et al., 2015 [27]	How GC affects the cancellous bone and the cellular events of the bone remodeling process	- Control (intact) OVX + deficient diet + GC	*n* = 20 9 ± 1 years	2 weeks after OVX, deficient diet and s.i. of methylprednisolone	7 mo after GC treatment	- Biochemistry: Serum CTX and OC; - Micro-CT: BV/TV, Tb.Th; - Histology; - IHC: TRAcP, SMA, Sp7, Runx2, ALP; - Histomorphometry: ES/BS, OS/BS, Oc.S/BS, Rv.S/BS	GC ↑ bone loss with ↓ BV/TV, Tb.Th, OC, OS/BS, cell density, Sp7, Runx2 and SMA; ↑ Rv.S/BS
Kiełbowicz et al., 2015a [28]	Changes in bone parameters in OVX and OVX + methylprednisol sheep in comparison to healthy sheep	- Control (intact) - OVX - OVX + methylprednisolone	*n* = 49 Merino 5–6 years	Methylprednisolone 30 days after OVX	21 days after the last application of steroidal medication (injections repeated 4 times at 20-day intervals)	Blood tests (estradiol, cortisol, progesterone, parathormone), diagnostic arthroscopy, micro-CT, and X-ray (BMD, BV/TV, BS/BV, porosity, Tb.N, Tb.Th, Tb. Sp, Conn.Dn)	↓ estrogens and progesterone levels, and ↑ parathormone and cortisol levels, OVX + methylprednisolone. ↓ bone turnover markers (b-ALP) in all groups. ↑ bone resorption markers (CTX), and ↓ radiological density in OVX and OVX + methylprednisolone groups
Kiełbowicz et al., 2015b [29]	Impact of steroidal medications on the structure and mechanical properties of OP animal model	- Control (intact) - OVX - OVX + glucocorticosteroid	*n* = 21 Merino 5–6 years	Treatment with GC (for 80 days) 1 mo after OVX	2 weeks after glucocorticosteroid administration	Quantitative CT	↓ radiological bone density in OVX + glucocorticosteroid group vs. control group
Kiełbowicz et al., 2016 [30]	Changes in bone parameters in OVX and OVX + methylprednisolone as opposed to parameters in healthy sheep	- Control (intact) - OVX - OVX + methylprednisolone	*n* = 49 Merino 5–6 years	Methylprednisolone 30 days after OVX	21 days after the last application of steroidal medication (injections repeated 4 times at 20-day intervals)	Mechanical tests (force/strength), SEM X-ray microanalysis (Ca/P), morphometric analysis (bone formation, porosity, thickness)	↑ bone formation, porosity and thickness, and ↓ Ca and P levels, strength, Young’s modulus, compressive strength of bone tissue, and deformation (strain) energy in OVX + methylprednisolone group
Kreipke et al., 2016 [31]	Effects of microarchitecture and estrogen depletion on microdamage susceptibility in trabecular bone	- Control (intact) - OVX	*n* = 19	2 years following OVX	2 years	Sequence of compressive and torsional overloads (propensity for microdamage formation in trabecular bone of the distal femur), mechanical testing, micro-CT (BMD, B.Ar., Dx.Ar., Cr.Dn., Cr.Ln., Cr.S.Dn.)	↓ BV/TV and ↑ SMI in the lateral condyle following OVX. ↓ Young’s modulus in OVX. ↓ Pre-existing Cr.Dn. with ↑ BV/TV in both Control and OVX, with a more negative slope in OVX. ↑ Cr.Dn. with ↑ SMI in OVX. Dx.Ar. correlated with ↑ SMI for OVX. In OVX Cr.Dn. from the compressive load correlated with preexisting Cr.Dn. as Dx.Ar., and both ↑ with ↓ BV/TV
Oheim et al., 2017 [32]	Effects of peripheral hormone therapy on centrally induced systemic bone loss	- Control (intact) - OVX - OVX + HPD; - OVX + HPD + estrogen - OVX + HPD + thyroxin - OVX + HPD + thyroxin + estrogen	*n* = 30 Corriedale 4–5 years	- OVX performed 1 week before HPD - Hormone therapy started 2 weeks after HPD	At the end of the 9-mo of hormone therapy	- Biochemistry urine (deoxypyridinoline) and serum parameters (sodium, potassium, chloride, Ca, P, Crea, BAP) - histology: iliac crest, spines (L3–L5) - X-rays, micro-CT, HR-pQCT: BV/TV, Tb.N, Tb.Sp, Tb.Th, Ct.Th - histomorphometry: BV/TV, Tb.N, Tb.Sp, Tb.Th - biomechanical test: femora three-point-bending test	Bone loss in OVX+HPD. Treatment with thyroxin alone ↑ bone resorption, ↓ bone volume. Treatment with estrogen and the combined treatment with estrogen and thyroxin prevent OVX-HPD-induced bone loss
Schulze et al., 2017 [33]	Selection of reference genes for quantitative real-time PCR	- Control (intact) - OVX - OVX + deficient diet (OVXD) - OVX + deficient diet + i.m. steroid-suspension (OVXDS)	*n* = 31 Merino 5.5 years	2 weeks after OVX diet and/or i.m steroid were given	8 mo after OVX	PCR L5 vertebra: GAPDH, ALAS1, HPRT, EF-2, G6PDH, ACTB, RPL19, B2M, YWHAZ, SDHA, PGK1, and TFRC	B2M, GAPDH, RPL19, and YWHAZ are the genes recommended for relative quantification of gene expression studies in ovine bone for evaluating bone-substituents in OP
Heiss et al., 2017 [34]	Clinically similar T-score standard to diagnose OP	- Control (sham) - OVX - OVX+deficient diet - OVX + deficient diet + GC	*n* = 32 Merino 3–9 years	2 weeks after OVX, deficient diet and i.m. injection of methylprednisolone	8 mo after GC	- DXA: BMD, BMC, and T-score and Z-score	OVX+GC ↓ LV, femur BMD, BMC. OVX+deficient diet ↓ LV, BMC, T-score during time
El Khassawna et al., 2017 [35]	Evaluation of RANKL/OPG *ratio* correlation to the method of OP induction	- Control (intact) (group 1) - OVX (group 2) - OVX + deficient diet (group 3) - OVX + deficient die t+ GC (group 4)	*n* = 32 Merino 5.5 years	2 weeks after OVX, deficient diet and i.m. injection of methylprednisolone	8 mo after GC	- DXA: BMD, Fat % for IC and LV; - Biomechanics: The maximum strength of the vertebrae, maximum load peak in the stiffness; - Histology; - IHC: ALP, TRAP, RANKL, OPG; - Histomorphometry: nonmineralized and mineralized matrix portion in IC and LV, Quantification of toluidine blue staining on LV, Spindle, spherical, and empty lacunae, RANKL or OPG activity; serum OCN, BAP, NTX; RT-PCR: ALP, CA2, OPG, RANKL, COL1A2, FN1 of L1	(Group 4): ↓ Z-score, maximum load at failure, stiffness, mineralized bone area, RANKL/OPG; ↑ OCN, empty lacunae, COL1A2, Fat% than the other groups. (Groups 3,4): ↓ NTX than (Groups 1,2). (Group 2): ↓ CA2 than the other groups. (Groups 2,4): ↓ OPG than (Groups 1,3)
Cabrera et al., 2018 [36]	Validation of the combination of OVX and GC	- Control (intact) - OVX - OVX + GC (2 and 5 mo)	*n* = 28 Merino 7–9 years	2 weeks after OVX, methylprednisolone administered for 2 or 5 mo	2 and 5 mo after OVX	- Biochemistry: serum OC, CTX-1; - DXA: Lumbar spine and femur BMC and aBMD. - pQCT: left tibia vBMD, bone area, BMC, cortical/subcortical and trabecular vBMD, cortical/subcortical and trabecular bone area, cortical/subcortical, and trabecular BMC	OVX + GC induces bone loss in a short period of time. OVX ↑ CTx-1 and OC. OVX + GC ↑ CTx-1 and ↓ OC at 5 mo. OVX and OVX + GC ↓ BMD, total and trabecular vBMD in the proximal tibia
Muller et al., 2019 [37]	Characterization of bone quality	- Control (sham) - OVX - OVX + deficient diet of Ca and vitamin D - OVX + deficient diet + GC injections	*n* = 28 5.5 years	NR	8 mo	Biomechanical testing and mathematical modelling, compression tests and finite-element analysis of stress states (stiffness, strength, averaged microscopic Young’s modulus at tissue level), micro-CT, and time-of-flight secondary ion mass spectrometry (trabecular structure, mineral and collagen distribution)	↓ BV/TV, Tb.Th, stiffness, strength and ↑ BS/BV in OVX + deficient diet + GC. ↑ Tb.Sp in OVX and OVX + deficient diet of Ca and vitamin D groups. ↓ Tb.N and ↑ SMI in OVX group
Cabrera et al., 2020 [38]	Effects of short- or long-term GC on plasma metabolites and lipids	- Control (intact) - OVX - OVX + GC (2 and 5 mo)	*n* = 28 Merino 7–9 years	2 weeks after OVX, methylprednisolone administered for 2 or 5 mo	2 and 5 mo after OVX	- Liquid chromatography–mass spectrometry untargeted metabolomic analysis	OVX + GC altered the metabolite and lipid profiles
Coelho et al., 2020 [39]	Characterization of GC-treated OVX sheep	- Control (sham) - OVX	*n* = 12 Portuguese Serra-da-Estrela 3–4 years	OVX + 1/week injections of dexamethasone (1 mg/kg) for 5 mo	6 mo after OVX	ELISA: serum TRAP, estradiol, BUN, Crea, TC, Ca, P, Mg, Glu, ALP activity, AST, ALT, GGT, TP; Micro-CT: BMD, BV/TV, BS/BV, Tb.Th, Tb.N, Tb.Sp, Po(cl), Po(op), Po(tot) of L4 vertebra body; Histology; Histomorphometry: Ct.Po % and Ct.Th were assessed in the cortical bone and the BV/TV, Tb.Th, Tb.Sp and Tb.N in the trabecular bone	OVX ↑ osteopenia, mean corpuscular volume, mean cell hemoglobin and monocytes, ALP, GGT, Mg, α1-globulin; ↓ red blood count and eosinophils, Crea, albumin, sodium, and estradiol
Rupp et al., 2021 [40]	Clinical relevance of a fracture defect model in the iliac crest	- Control (intact) - OVX - OVX + deficient diet (OVXD) - OVX + deficient diet + i.m. steroid-suspension (OVXDS)	*n* = 31 Merino 5.5 years	2 weeks after OVX diet and/or i.m steroid were given, and iliac crest bone defects (7.5 mm diameter and 25 mm long) were created	5 and 8 mo after OVX	- DXA: BMD, BMC - Micro-CT - Histology and histomorphometry: Tb.Th, Ct.Th, size callus formation, cartilage tissue, mineral deposition rate	Significant ↓ BMD and BMC in OVXDS. OVX and OVXD showed complete healing after 8 mo. Bone quality affected mostly in the OVXDS group with irregular trabecular network, ↓ Ct.Th, ↑ cartilaginous tissue at 8 mo. The mineral deposition rate showed a declining pattern in the control, OVX, and OVXD from 5 mo to 8 mo. OVXDS group showed an incremental pattern from 5 mo to 8 mo

**Abbreviations:** Ref (Ref.), increase (↑), decrease (↓), ovariectomy (OVX), number (n), not reported (NR), months (mo), microcomputed tomography (micro-CT), reverse transcriptase-polymerase chain reaction (RT-PCR), receptor activator of nuclear factor kappa-B ligand (RANKL), osteoprotegerin (OPG), collagen type I alpha 1 chain (COL1A1), collagen type I alpha 2 chain (COL1A2), osteocalcin (OCN), osteopontin (OPN), Fourier-transform infrared spectroscopy (FTIR), osteoporosis (OP), hypothalamo pituitary disconnection (HPD), low elastic modulus expandable pedicle screw (L-EPS), expandable pedicle screws (EPS), bisphosphonate-loaded calcium phosphate cement (BP-loaded CaP cement), bone volume (BV/TV, %), trabecular number (Tb.N, μm), trabecular spacing (Tb.Sp, μm), trabecular thickness (Tb.Th, μm), osteoblast number (N.Ob/B.Pm), osteoclast number (N.Oc/B.Pm), surface indices (Ob.S/BS and Oc.S/BS), eroded surface (ES/BS), osteoid surface (OS/BS), mineralized surface (MS/BS), bone formation rate (BFR/BS mm^3^/mm^2^/year), cortical thickness (Ct.Th, μm), bone mineral density (BMD), bone mineral density distribution (BMDD), bisphosphonate (BP), calcium phosphate (CaP), calcium (Ca), trabecular bone pattern factor (TbPf, mm-1), trabecular area/tissue area (Tb.Ar/T.Ar), bone mineral content (BMC), scanning electron microscopy (SEM), high-resolution peripheral quantitative computed tomography (HR-pQCT), quantitative backscattered electron imaging (qBEI), immunohistochemistry (IHC), alkaline phosphatase (ALP), human bone alkaline phosphatase (b-ALP), runt-related transcription factor 2 (Runx2), tartrate-resistant acid phosphatase (TRAP), enzyme-linked immunosorbent assay (ELISA), dual-energy X-ray absorptiometry (DXA), tissue mineral density (TMD), degree of anisotropy (DA), reversal surface (Rv.S/BS), bone area (B.Ar.), diffuse damage area (Dx.Ar.), crack density (Cr.Dn.), crack length (Cr.Ln.), crack surface density (Cr.S.Dn.), structural model index (SMI), bone-specific alkaline phosphatase (BAP), N-terminal telopeptide (NTX), new bone formed per total bone defect area (NB/TA), glucocorticoid (GC), blood urea nitrogen (BUN), creatinine (Crea), total proteins (TP), magnesium(Mg), gamma-glutamyl transferase (GGT), aspartate aminotransferase (AST), alanine aminotransferase (ALT), glucose (Glu), total cholesterol (TC), phosphorus (P), osterix (Sp7), spinal muscular atrophy (SMA).

## 3. Results

### 3.1. Study Selection and Characteristics

The initial literature search retrieved 123 studies. Of those, 38 studies were identified using PubMed, 46 using Scopus, and 39 were found in Web of Science Core Collection. Articles were screened for title and abstract, and 78 articles were selected. Subsequently, these articles were submitted to a public reference manager to eliminate duplicates, and 44 complete articles were then reviewed to establish whether the publications met the inclusion criteria. Three full-text articles were not found, 21 articles were excluded because the absence of a sham operated or no operated control group, and 2 articles were excluded because of involved biomaterials use; finally, 18 studies were considered eligible for this review. Search strategy and study inclusion and exclusion criteria are detailed in Figure 3.

### 3.2. Assessment of Methodological Quality

Assessment of methodological quality for each study was summarized in Table 3. The 61% of the studies were rated strong, 33% were rated moderate, and 6% were rated weak. Methodological weaknesses that led to moderate or weak quality scores prevalently included: Study design, ethical statement, experimental procedure, experimental animals, housing and keeping, sample size, allocation of animals to experimental groups, experimental outcomes, statistical methods (for the section Methods), baseline data, numbers analyzed, outcomes and estimation, adverse events (for the section Results), interpretation and scientific implications, generalizability and translation, and funding (for the section Discussion).

### 3.3. Studies Results

#### 3.3.1. General Characteristics: Breed, Age, Surgical Methods for OVX

Although there is no head-to-head study to compare the effects of OVX on bone parameters in different sheep breeds, Merino sheep were the most used breed (50%) in this review. Six studies did not report the sheep breed (33%) while the remaining used Corriedale breed (11%) and Portuguese Serra-da-Estrela breed (6%). All the studies considered animals that reached sexual maturity. However, it is important to underline that at sexual maturity, the sheep is not mature from the skeletal point of view yet, and growth ends a long time after puberty, such as in the vertebral body epiphysis where the closing times of the ossification centers are around 4–5 years of age. In this review, the age of the animals ranged from 3 to 10 years with some studies using animals with a wide age range [34,36,38]. The age reported in all studies are those at which the animals underwent bilateral OVX that was always performed through a ventral midline laparotomy under general anesthesia (Figure 4). All animals had at least one-week acclimatization period prior to the OVX. General anesthesia was prevalently induced by intravenous injection of ketamine and propofol and prolonged with isoflurane in oxygen through an endotracheal tube [23,24,25,26,27,28,29,30,31,32,33,34,35,36,37,38,39,40]. Antibiotic prophylaxis was provided by intravenous injection at the time of anesthesia induction. After OVX, animals were fed a maintenance ratio of hay and had water ad libitum.

#### 3.3.2. Control Group

All the included studies had a control group in the 83% that was represented by unoperated animals while a total of 17% was represented by the sham group, i.e., a group where the procedure mimics the OVX in every way, including preprocedural routine, anesthesia, incisions, and postprocedural follow-up. Sham procedures are generally considered a more appropriate control than using no intervention since this procedure more clearly distinguishes whether a new treatment is effective beyond the placebo response that is due to the contextual cues of an invasive procedure [39]. However, from an ethical point of view, unmanipulated (unoperated) control animals minimize animal distress and unnecessary procedures.

#### 3.3.3. Verification of OVX

The success of OVX in sheep was mostly confirmed by iliac crest and/or lumbar vertebra microcomputed tomography (CT) analyses [23,25,26,27,31,32,39,40] as well as by quantitative computed tomography (qCT), which measured the BMD using a X-ray CT scanner with a calibration standard to convert Hounsfield units (HU) of the CT image to bone mineral density values [24,28,29]. BMD was also measured by dual-energy X-ray absorptiometry (DXA) in iliac crest, femora, and lumbar spine [25,34,35,36,37,40]. Same studies also evaluated OVX through analysis of biochemical markers of bone, such as osteocalcin, C-terminal cross-linked telopeptides of type I collagen (CTXI), alkaline phosphatase (b-ALP), and by changes in hormonal profile (circulating concentrations of estradiol) [27,28,36,38,39].

#### 3.3.4. Osteoporosis Induced by OVX Alone

In 21.3% of the included studies [23,25,26,31], the primary outcome was to evaluate specifical changes in microarchitectural and mechanical parameters between unoperated and OVX animals in proximal femurs [23,25,26,31], lumbar vertebra [25,26], mandibular angle, and rib [25] (Figure 5). Results showed a reduction mineral-to-matrix ratio, ratio of mature to immature collagen cross-linking, trabecular number (Tb.N), trabecular thickness (Tb.Th) and compressive strength, and an increase in trabecular separation (Tb.Sp) starting from 12 months after OVX [23,25,26,31]. Analyzing different anatomical sites in lumbar vertebra, the microarchitectural characteristics were shown to be more significantly degraded in comparison to femoral neck and to ribs [31]. These results were also confirmed by other studies [25,26] 12 months after OVX that showed a reduction in bone volume by 45.6%, 36.1%, 21.3%, and 18.7% in lumbar vertebrae, femoral necks, mandibular angles, and ribs, as well as in trabecular area/tissue area (Tb.Ar/T.Ar) by 32.1%, 23.2%, and 20.7% in lumbar vertebrae, femoral necks, and mandibular angles, confirming that vertebral body is the preferred anatomic site for studying bone from the OVX sheep models [25]. Furthermore, Kreipke et al. also showed that the mechanical anisotropy, determined from microscale finite element models, was altered and greater in the OVX group than in the control group [26].

#### 3.3.5. Osteoporosis Induced by OVX Associated to Steroid Therapy and/or Deficient Diet

The 67.6% of the included studies evaluated how steroid therapy (methylprednisolone or dexamethasone) and/or deficient diet (of calcium and/or Vitamin D) in combination with OVX affect and/or impact the bone remodeling process and/or bone quality (Figure 5). In detail, eight studies reported the combination of deficient diet and steroid therapy to OVX while four studies reported the association of steroid therapy and OVX. Except for two studies [37,39] where initiation of steroid therapy or deficient diet is not specified, all other studies begin at two (70%) or four weeks (30%) after OVX. The experimental time varies from three to eight months after OVX. Three months after OVX and methylprednisolone therapy (4 injections at 20-day intervals), a reduction in bone turnover markers, e.g., b-ALP, in bone density, and an increase in bone resorption markers, e.g., CTXI, were seen both in OVX and OVX with methylprednisolone therapy [28,29]. Contrastingly, a reduction in estrogens and progesterone levels and an increase in parathormone and cortisol levels were detected only in OVX with methylprednisolone therapy [28]. Similarly, a reduction in calcium and phosphorus levels, strength, Young’s modulus, compressive strength of bone tissue, and deformation (strain) energy was present in OVX with methylprednisolone group [30]. After five months, OVX with methylprednisolone was shown to induce bone loss in a short period of time in comparison to OVX alone, increasing CTXI and reducing osteocalcin (OC) [36]. Furthermore, at this experimental time, OVX with methylprednisolone was also shown to alter the metabolic and lipid profile in comparison to OVX alone [38]. However, BMD reduction was similar between OVX and OVX with methylprednisolone [36]. At six months, OVX associated with weekly dexamethasone injection led to an increase in mean corpuscular volume, mean cell hemoglobin and monocytes, alkaline phosphatase, gamma-glutamyl transpeptidase, magnesium, and α1-globulin as well as a decrease in red blood count and eosinophils, creatinine, albumin, sodium, and estradiol in comparison to sham-control animals [39]. At seven months, OVX associated to deficient diet and methylprednisolone injections revealed a reduction in bone volume/total volume (BV/TV), Tb.Th, OC, (osteoid surface/bone surface) OS/BS, cell density, osterix (Sp7), runt-related transcription factor 2 (Runx2) and spinal muscular atrophy (SMA) in comparison to control animals [27]. El Ehassawna et al. [35] showed that after eight months of methylprednisolone injections and dietary treatment (deficient calcium and/or vitamin D diet) in OVX sheep the global receptor activator of nuclear factor kappa-B ligand/osteoprotegerin (RANKL/OPG) ratio was not different from that of the control. Interestingly, assessment of the osteocyte-specific RANKL/OPG ratio showed that the steroid-induced osteoporosis in its late progressive phase stimulates RANKL expression in osteocytes [35]. Constrastingly, Muller et al. reported a reduction in bone volume, Tb.Th, stiffness, and strength, as well as an increase in bone surface/bone volume (BS/BV) in OVX with deficient diet and glucocorticoid in comparison to OVX alone group or OVX with special diet deficient of calcium and vitamin D group [37]. Similar results were also found by Heiss et al. [34]. Furthermore, a study by Rupp et al. investigated the clinical relevance of a fracture defect model in the iliac crest of OVX animals in combination with diet deficiency and steroid administration, highlighting that this model showed significantly lower bone mineral density and bone mineral content (BMC) at all time points (five and eight months after OVX) in comparison to unoperated control, OVX group, and OVX with deficient diet treatment [40]. Despite OVX with deficient diet, glucocorticoid proved to be the most suitable osteoporosis model in sheep, and it was recommended to test B2M, GAPDH, RPL19, and YWHAZ gene expression that represented standard reference genes for a relative quantification of transcriptional activity of ovine bone [33].

#### 3.3.6. Osteoporosis Induced by OVX and Hypothalamic–Pituitary Disconnection

Two (11.1%) studies evaluated the OVX effects associated to hypothalamic–pituitary disconnection (HPD), a centrally induced osteoporosis model, obtained through surgical disconnection of the hypothalamus and pituitary gland [24,32] (Figure 5). To determine the sustainability of bone loss and its biomechanical relevance, HPD in OVX sheep were studied 24 months after surgery in comparison to untreated control, OVX alone, and OVX sheep 12 months after surgery [24]. Twenty-four months after HPD, histomorphometrical analyses of the iliac crest showed a significant reduction in BV/TV by 60% in comparison to control. Cortical thickness of the femora measured by high-resolution peripheral pqCT did not change between 12 and 24 months after HPD but remained decreased by 30% [24]. Subsequently, to investigate the role of peripheral hormones in this centrally induced systemic bone loss model, a hormone replacement experiment was planned [32]. Therefore, estrogen (OHE), thyroxin (OHT), or a combination of both (OHTE) was substituted in OVX HPD sheep and compared to untreated sheep, OVX, and OVX + HPD sheep. Results showed that peripheral hormone substitution prevented HPD-induced low-turnover osteoporosis in sheep [32].

## 4. Discussion

Although there is no perfect animal model, the relationships between the human and sheep skeleton, in response to estrogen deficiency and specific therapeutic agents, made OVX sheep a suitable in vivo model for osteoporosis research. As highlighted also by this review, the advantages related to sheep use are many: (1) the skeleton size and mechanical features of are similar to humans [38,39,42,43,44,45,46]; (2) older sheep show Haversian bone remodeling [38]; (3) they are genetically more human-like than rodents; (4) sheep show sex hormone profiles comparable to women; (5) Merino sheep have an almost continuous the oestrus cycle [40]; (6) they are relatively easy to handle and reasonable to maintain [42]; and (7) they present fewer ethical concerns than other nonhuman primates [42]. However, several disadvantages are also present. One of these is represented by their different gastrointestinal system; therefore, studies in which there is interest in the oral absorption of specific therapies and/or drugs should involve the surgical insertion of an abomasal fistula or the stimulation of the oropharynx during administration [42]. Another potentially relevant difference in the mineral homeostasis between sheep and humans is in their phosphorus metabolism, since the urinary phosphate excretion is much lower in sheep and the gastrointestinal tract is the major route of elimination [40]. Another key aspect is that the bone mineral density and bone mineral content are significantly higher in sheep compared to humans leading to an even more pronounced increase in mechanical stability [43]. Finally, since bone mineral density and bone turnover parameters change considerably in sheep during the years, appropriate control groups are always essential [44].

Although in this review we found a small number of studies where a control group was present (*n* = 18), all of them documented the response of the sheep bones to OVX in comparison to non-operated and/or sham animals. In fact, starting from 12 months after OVX, a significant decline in bone mineral density and microarchitectural bone parameters, prevalently trabecular bone volume, trabecular thickness, number, and separation, in lumbar vertebrae, iliac crest, and femora as well as in biochemical markers, were detected in almost all the studies. This review also showed that the sensibility of OVX at this time point was site-specific, with major alteration in lumbar spine followed by femoral neck [21]. Thus, the influence of OVX on bone metabolism in sheep seems to be comparable to post-menopausal osteoporosis in humans. Before 12 months from OVX, i.e., starting from 5 months, microarchitectural bone changes and biochemical marker alterations were present when osteoporosis was induced by OVX associated to steroid therapy, prevalently methylprednisolone, and/or deficient diet. OVX associated to steroid therapy was used in many of the included studies (more than 66%). In comparison to OVX alone, this model is certainly easier and more reliable for the induction of pronounced cortical and trabecular bone loss [45,46]. However, a major disadvantage of this model is the requirement for prolonged glucocorticoid injections to achieve the bone loss that can compromise animal welfare with severe side effects, such as massive infections and hair loss [45,46]. Few studies also evaluated osteoporosis induced by OVX associated to surgical disconnection of the hypothalamo–pituitary axis that led to a profound bone loss in cortical as well as trabecular bone. However, this model can lead to several systemic alterations because of blood level changes in different hormones [47].

A limitation of almost all the included studies, regardless of whether OVX alone or associated with other treatments, was related to the low number of animals included in the research. On the one hand, it is mandatory to minimize the number of animals used to be consistent with scientific aims; on the other hand, the 3Rs principle (replacement, reduction, and refinement) also underline that it is mandatory to have robust and reproducible animal experiments. This limit is particularly evident from the results of methodological quality where many studies showed an inadequate sample size. Probably the small number of animals used in the studies can also be linked to the low number of studies found in this review where the presence of a control group was one of the main inclusion criteria. Obviously, the use of a control group requires the use of an even greater number of animals; this would allow us to have a greater number of reproducible, robust, and translatable data. In fact, despite OVX, sheep models seem to be a well-established animal models for osteoporosis. Some important areas, however, remain to be elucidated regarding this model, and the studies’ results should be constantly interpreted against the background of strain, age, season, diet, skeletal site, and hormone-cycle features, consequently, appropriate control groups are crucial.

## 5. Conclusions

In conclusion, this review highlighted that OVX can influence the bone metabolism in sheep causing a pronounced systemic bone loss and structural deterioration/alteration comparable to the situation found in osteoporosis patients. Future studies are mandatory to clarify specific aspect of this model, including molecular mechanisms involved in OVX-induced osteoporosis and the impact of OVX on the different gastrointestinal system present in sheep. Despite these limits and the restricted number of studies found for this review, it is important to underline that data for treating osteoporosis patients are based not only on good planning and study design but also on correct animal use and deep knowledge of specific differences between the healthy and OVX animals.

## Figures and Tables

**Figure 1 ijms-23-08904-f001:**
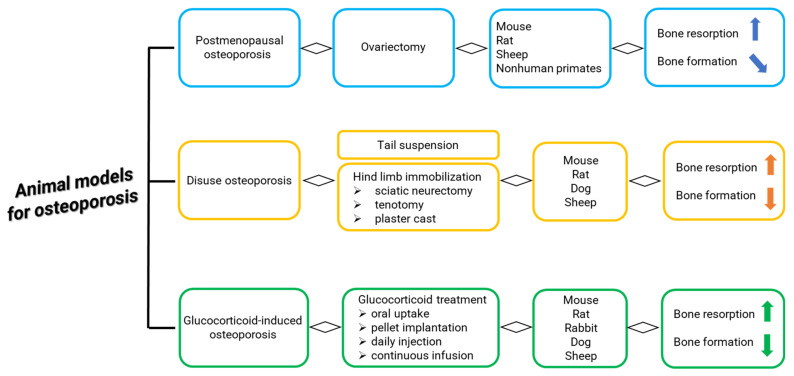
Main methods and animals for generation of (1) postmenopausal osteoporosis, (2) disuse osteoporosis, and (3) glucocorticoid-induced osteoporosis [5,6,7].

**Figure 2 ijms-23-08904-f002:**
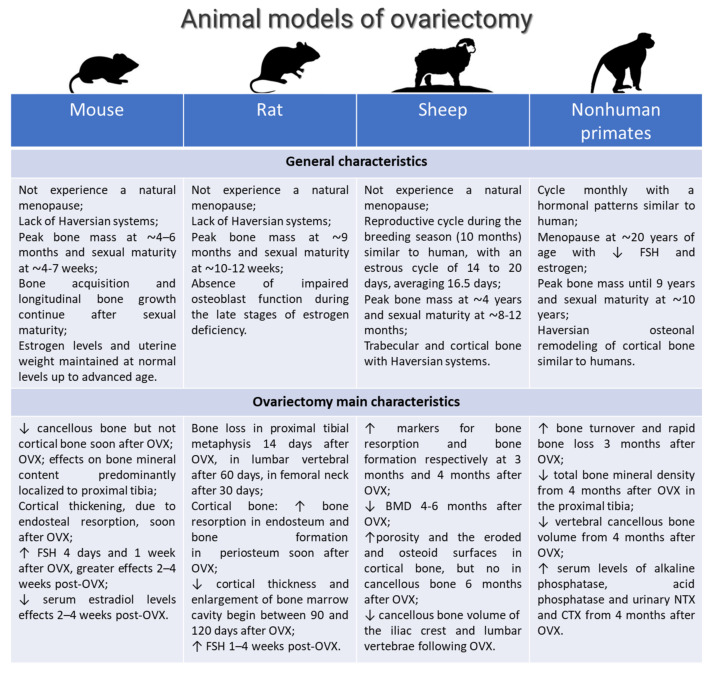
Main characteristics and principal OVX features of the most common animal models of postmenopausal osteoporosis [5,6,7,9,10,11,12,13,14,15,16,17,18,19,20]. ↑: increase; ↓: decrease.

**Figure 3 ijms-23-08904-f003:**
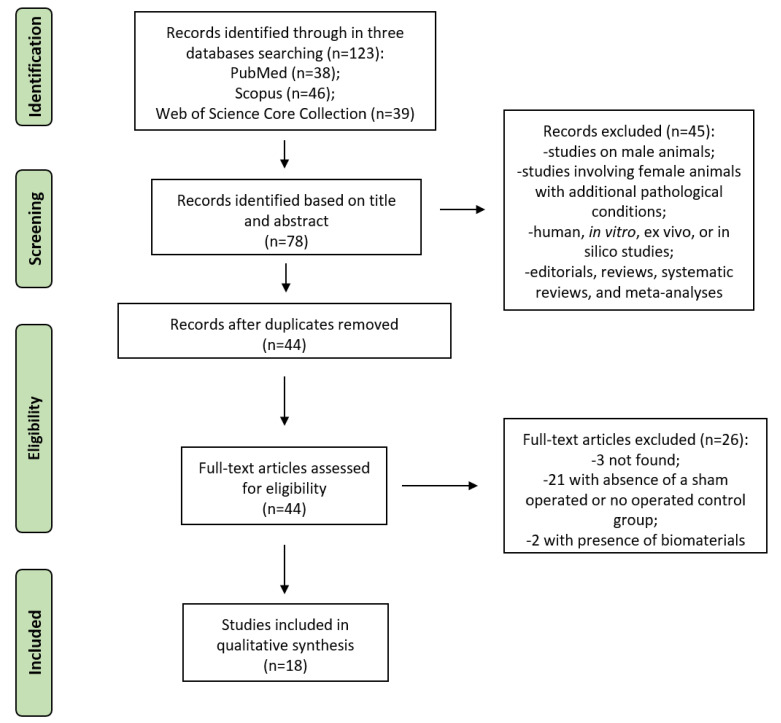
Systematic review flow diagram. The PRISMA flow diagram for the systematic review detailing the database searches, the number of abstracts screened, and the full texts retrieved.

**Figure 4 ijms-23-08904-f004:**
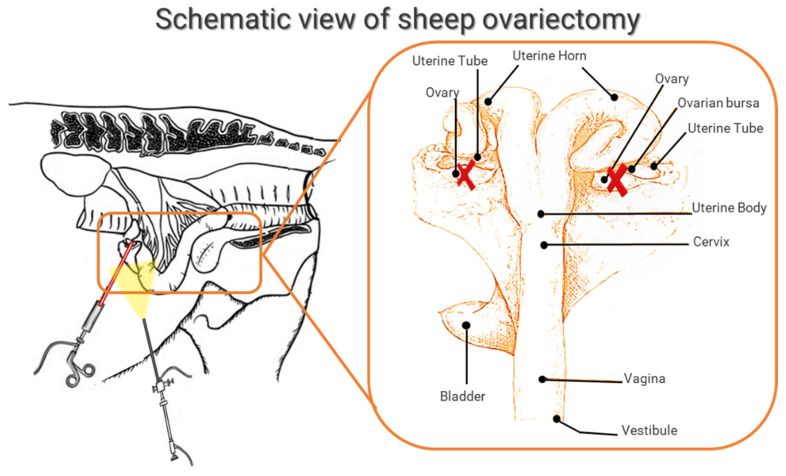
Schematic representation of bilateral OVX in sheep performed through a ventral midline laparotomy [41]. X: ovary excision.

**Figure 5 ijms-23-08904-f005:**
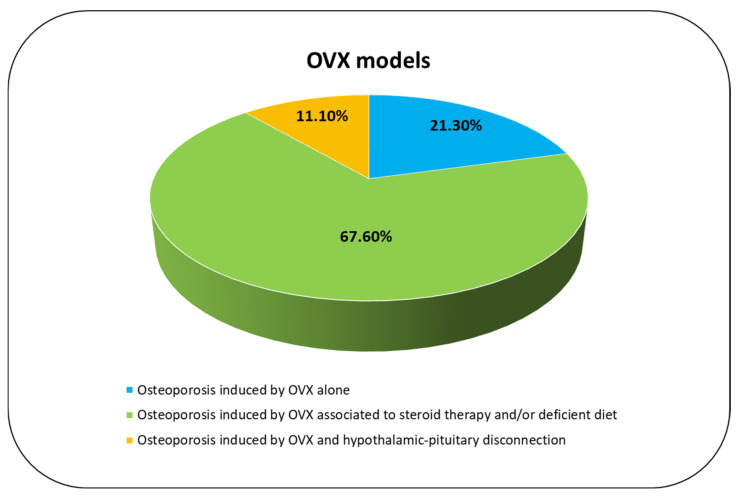
OVX models analyzed in this review.

**Table 1 ijms-23-08904-t001:** Search terms used in the PubMed, Scopus, and Web of Science Core Collection.

Database	Free-Vocabulary and/or Medical Subject Headings (MeSH) Terms
**PubMed**	(“ovariectomied” [All Fields] OR “ovariectomy” [MeSH Terms] OR “ovariectomy” [All Fields] OR “ovariectomies” [All Fields]) AND (“sheep” [MeSH Terms] OR “sheep” [All Fields] OR “sheeps” [All Fields] OR “sheep s” [All Fields] OR “sheep, domestic” [MeSH Terms] OR (“sheep” [All Fields] AND “domestic” [All Fields]) OR “domestic sheep” [All Fields]) AND (“bone and bones” [MeSH Terms] OR (“bone” [All Fields] AND “bones” [All Fields]) OR “bone and bones” [All Fields] OR “bone” [All Fields]) AND ((y_10[Filter]) AND (english[Filter]))
**Web of Science Core Collection**	(TS = ovariectomied OR TS = ovariectomy) AND (TS = sheep OR TS = sheep, domestic) AND (TS = bone and OR TS = bones)—With Publication Year from 2012 to 2022
**Scopus**	(TITLE-ABS-KEY (ovariectomy) OR TITLE-ABS-KEY (ovariectomied) AND TITLE-ABS-KEY (sheep) OR TITLE-ABS-KEY (sheep, domestic) OR TITLE-ABS-KEY (bone AND bones)) AND PUBYEAR > 2012

**Table 3 ijms-23-08904-t003:** Categories to assess the quality of finally selected studies.

Ref.	Title	Abstract	Introduction	Methods	Results	Discussion	Total
Brennan et al., 2012 [23]	0	1	3	14	4	5	27
Oheim et al., 2013 [24]	1	1	3	8	3	4	20
Zhang et al., 2014 [25]	1	2	3	6	5	3	20
Kreipke et al., 2014 [26]	1	1	3	11	3	2	21
Andreasen et al., 2015 [27]	1	2	3	16	7	4	33
Kiełbowicz et al., 2015a [28]	1	1	2	12	4	3	23
Kiełbowicz et al., 2015b [29]	1	1	3	10	3	2	20
Kiełbowicz et al., 2016 [30]	1	1	2	12	4	3	23
Kreipke et al., 2016 [31]	0	1	3	7	2	4	17
Oheim et al., 2017 [32]	1	1	3	8	6	5	24
Schulze et al., 2017 [33]	1	2	3	8	3	3	20
Heiss et al., 2017 [34]	1	2	3	13	3	3	25
El Khassawna et al., 2017 [35]	1	2	3	14	3	1	24
Cabrera et al., 2018 [36]	1	2	3	15	3	6	30
Muller et al., 2019 [37]	1	2	3	13	3	3	25
Cabrera et al., 2020 [38]	1	2	3	15	4	5	30
Coelho et al., 2020 [39]	1	2	3	14	3	4	27
Rupp et al., 2021 [40]	1	2	3	8	6	6	25

The quality assessment includes 20 items: Title (1), abstract/summary (2), introduction/primary and secondary objectives, methods/study design (4), methods/ethical statement (5), methods/study design (6), methods/experimental procedure (7), methods/experimental animals (8), methods/housing and keeping (9), methods/sample size (10), methods/allocation animals to experimental groups (11), methods/experimental outcomes (12), methods/statistical methods (13), results/baseline data (14), results/numbers analyzed (15), results/outcomes and estimation (16), results/adverse events (17), discussion/interpretation and scientific implications (18), discussion/generalizability and translation (19), discussion/funding (20). predefined gradings [i.e., 0 = inaccurate—not concise (item 1)/clearly inaccurate (item 2)/clearly insufficient (items 3, 5–9)/not clear (item 4)/no (items 11–14, 16, 17)/clearly inadequate (items 10, 15, 18–20); 1 = accurate and concise (item 1)/possibly accurate (item 2)/possibly sufficient (items 3, 5–9)/clear (items 4)/possibly adequate (items 10, 15, 18–20)/yes (items 11 and 14)/unclear or not complete (items 12, 13, 16, 17); 2 = clearly accurate (item 2)/clearly sufficient (items 3, 5–9)/clearly adequate (items 10, 15, 18–20)/yes (items 12, 13, 16, 17)] were applied for the different items.

## Data Availability

Not applicable.

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
