# Peer review of "Osteoporosis Preclinical Research: A Systematic Review on Comparative Studies Using Ovariectomized Sheep"

_ijms, 2022, doi:10.3390/ijms23168904_

Round 1

Reviewer 1 Report

This review manuscript should provide more figures (recommend 3-5). Such as, the process of sheep ovariectomy, and what's the difference from other animals (rat, mouse, etc.). 

Author Response

This review manuscript should provide more figures (recommend 3-5). Such as, the process of sheep ovariectomy, and what's the difference from other animals (rat, mouse, etc.). 

As suggested by the reviewer we added 4 figures.

In detail, Figure 1 represents the main animal models described in literature for osteoporosis, i.e. postmenopausal osteoporosis, disuse osteoporosis, and glucocorticoid-induced osteoporosis. Figure 2 shows main characteristics of postmenopausal OP induced by ovariectomy in mouse, rat, sheep, and nonhuman primates. Figure 4 is a schematic representation of bilateral OVX in sheep performed through a ventral midline laparotomy. Finally, Figure 5 summarizes the number and type OVX models used in this review, i.e. OVX alone, OVX associated to steroid therapy and/or deficient diet and OVX associated to hypothalamic-pituitary disconnection.

Reviewer 2 Report

The systematic review by Salamanna et al. investigates the use of sheep subjected to OVX as a model for post-menopausal osteoporosis. The manuscript is interesting, well-written, and the discussion is sensible. Below a few specific minor comments to improve the manuscript.

Minor comments:

·      Abstract: Commas would be useful in this sentence: ”Sheep ovariectomy (OVX) alone or associated to steroid therapy or deficient diet or hypothalamic-pituitary disconnection have proven to be of critical importance for osteoporosis (OP) research in orthopedic.”

·      General: To increase the readability, the abbreviation OP should be omitted.

·      Introduction: “Based on literature data, sheep have proven invaluable in orthopedic research and specifically for OP research [6,7].” It could also be emphasized that sheep are also used to serve as other animal models of bone loss such as from disuse (see the systematic review by Brent et al. https://link.springer.com/article/10.1007/s00223-020-00799-9).

·      General: “in vitro, ex vivo, or in silico” check the manuscript to ensure all instances are in italic.

·      “Free words, and controlled vocabulary specific to each bibliographic database were com- 96 bined using the operator “OR”. Please specify “free words and controlled vocabulary” and provide the complete search string for all databases searched either under “2.2. Search strategies” or as an appendix.

·      “3.2. Assessment of methodological quality” Please provide more details on how the scores were assessed. In the present form, it is a bit confusing what merit different scores in Table 2.

·      General: There are a few typos throughout the manuscript that should be addressed. Some abbreviations are introduced twice (such as CTXI) – please check the manuscript for duplicates.

Author Response

The systematic review by Salamanna et al. investigates the use of sheep subjected to OVX as a model for post-menopausal osteoporosis. The manuscript is interesting, well-written, and the discussion is sensible. Below a few specific minor comments to improve the manuscript.

Minor comments:

-Abstract: Commas would be useful in this sentence: ”Sheep ovariectomy (OVX) alone or associated to steroid therapy or deficient diet or hypothalamic-pituitary disconnection have proven to be of critical importance for osteoporosis (OP) research in orthopedic.”

As suggested, we added commas in this sentence.

-General: To increase the readability, the abbreviation OP should be omitted.

As suggested, we omitted the OP abbreviation.

-Introduction: “Based on literature data, sheep have proven invaluable in orthopedic research and specifically for OP research [6,7].” It could also be emphasized that sheep are also used to serve as other animal models of bone loss such as from disuse (see the systematic review by Brent et al. https://link.springer.com/article/10.1007/s00223-020-00799-9).

As suggested, we highlighted this aspect in the introduction section (Page 2, Lines 47-49) and in Figure 1. In addition, we added the reference suggested by the reviewer (reference 5).

-General: “in vitro, ex vivo, or in silico” check the manuscript to ensure all instances are in italic.

We checked and changed all “in vitro, ex vivo, or in silico” in italic.

-“Free words, and controlled vocabulary specific to each bibliographic database were com- 96 bined using the operator “OR”. Please specify “free words and controlled vocabulary” and provide the complete search string for all databases searched either under “2.2. Search strategies” or as an appendix.

As suggested, we added a table (Table 1) where we reported the complete search string for all databases used in this review.

-“3.2. Assessment of methodological quality” Please provide more details on how the scores were assessed. In the present form, it is a bit confusing what merit different scores in Table 2.

As suggested, we specified in detail how the score was assessed. We have put the methodological quality assessment specifications under the table 3 in the manuscript. “The score includes 20 items: Title (1), Abstract/Summary (2), Introduction/Primary and secondary objectives, Methods/Study design (4), Methods/Ethical statement (5), Methods/Study design (6), Methods/ Experimental procedure (7), Methods/Experimental Animals (8), Methods/Housing and keeping (9), Methods/Sample size (10), Methods/ Allocation animals to experimental groups (11), Methods/Experimental outcomes (12), Methods/Statistical methods (13), Results/Baseline data (14), Results/Numbers analyzed (15), Results/Outcomes and estimation (16), Results/Adverse events (17), Discussion/Interpretation and scientific implications (18), Discussion/ Generalizability and translation (19), Discussion/Funding (20). Predefined gradings [i.e. 0 = inaccurate – not concise (item 1)/clearly inaccurate (item 2)/clearly insufficient (items 3, 5 –9)/not clear (item 4)/no (items 11, 12 –14, 16, 17)/clearly inadequate (items 10, 15, 18–20); 1 = accurate and concise (item 1)/possibly accurate (item 2)/possibly sufficient (items 3, 5–9)/ clear (items 4)/possibly adequate (items 10, 15, 18–20)/yes (items 11 and 14)/unclear or not complete (items 12, 13, 16, 17); 2 = clearly accurate (item 2)/clearly sufficient (items 3, 5– 9)/clearly adequate (items 10, 15, 18– 20)/yes (items 12, 13, 16, 17)] were applied for the different items.”

-General: There are a few typos throughout the manuscript that should be addressed. Some abbreviations are introduced twice (such as CTXI) – please check the manuscript for duplicates.

We check and eliminate typos throughout the manuscript.

Round 2

Reviewer 1 Report

The revised manuscript has addressed most of review's concerns, and can be considered publication on IJMS.